# Genome-Wide Identification of Genes Encoding for Rho-Related Proteins in ‘*Duli*’ Pear (*Pyrus betulifolia* Bunge) and Their Expression Analysis in Response to Abiotic Stress

**DOI:** 10.3390/plants11121608

**Published:** 2022-06-19

**Authors:** Gang Li, Pingli Song, Xiang Wang, Qingcui Ma, Jianfeng Xu, Yuxing Zhang, Baoxiu Qi

**Affiliations:** 1Hebei Pear Engineering Technology Research Center, College of Horticulture, Hebei Agricultural University, Baoding 071001, China; liganghebau@163.com (G.L.); pinglisong2022@163.com (P.S.); qq384986509@163.com (X.W.); maqingcui2020@163.com (Q.M.); xjf@hebau.edu.cn (J.X.); 2School of Pharmacy and Biomolecular Science, Liverpool John Moors University, Liverpool L3 3AF, UK

**Keywords:** ‘*Duli*’ pear, *Pyrus betulifolia* bunge, rho-related proteins, ROP, abiotic stress

## Abstract

Twelve Rho-related proteins (ROPs), namely PbROPs, were identified from the genome of the recently sequenced ‘*Duli*’ pear (*Pyrus betulifolia* Bunge), a wild-type pear variety routinely used for rootstocks in grafting in China. The length and molecular weight of these proteins are between 175 and 215 amino acids and 19.46 and 23.45 kDa, respectively. The 12 *PbROPs* are distributed on 8 of the 17 chromosomes, where chromosome 15 has the highest number of 3 *PbROPs*. Analysis of the deduced protein sequences showed that they are relatively conserved and all have the G domain, insertion sequence, and HVR motif. The expression profiles were monitored by quantitative RT-PCR, which showed that these 12 *PbROP* genes were ubiquitously expressed, indicating their involvement in growth and development throughout the life cycle of ‘*Duli*’ pear. However, they were altered upon treatments with abscisic acid (ABA, mimicking abiotic stress), polyethylene glycol (PEG, mimicking drought), and sodium chloride (NaCl, mimicking salt) to tissue-cultured seedlings. Further, transgenic Arabidopsis expressing *PbROP1*, *PbROP2*, and *PbROP9* exhibited enhanced sensitivity to ABA, demonstrating that these 3 *PbROPs* may play important roles in the abiotic stress of ‘*Duli*’ pear. The combined results showed that the ‘*Duli*’ genome encodes 12 typical ROPs and they appeared to play important roles in growth, development, and abiotic stress. These preliminary data may guide future research into the molecular mechanisms of these 12 PbROPs and their utility in molecular breeding for abiotic stress-resistant ‘*Duli*’ pear rootstocks.

## 1. Introduction

ROPs (Rho-related proteins from plants) are GTPases of the small monomeric G proteins that are unique to plants. They were first discovered in 1993 in the pea plant [1]. Since then, many more such proteins have been identified from different plant species, including 11 from Arabidopsis [2,3], 7 from rice [4], 9 from maize [5], 22 from tobacco [6], and 17 from banana [7]. Studies have shown that these small GTPases play vital roles in cell division and differentiation, pollen tube growth, root hair formation, response to hormones, drought, salt, and diseases [7,8].

ROP GTPase functions as a ‘molecular switch’ between its two exchangeable active GTP binding and inactive GDP binding forms. Three proteins can regulate these two forms of ROP GTPases, the guanine nucleotide exchange factor (GEF), the GTPase-activating protein (GAP), and the guanine nucleotide dissociation inhibitor (GDI). The conversion from the inactive GDP- to the active GTP-binding form is catalyzed by GEF. The activated ROP-GTP can interact with downstream effectors to initiate specific signaling pathways. When the signaling process finishes, GAP, the negative regulator, can promote the hydrolyzation of GTP to GDP and return the active ROP-GTP to its inactive form ROP-GDP, which is ready for the next round of signaling processes [9].

ROP proteins contain three highly conserved domains, the G domain (G1–G5), the insertion sequence between G4 and G5, and the hyper variable region (HVR) in the C-termini [10,11]. The G domain has five highly conserved regions G1–G5, which are shared within all the small G protein families. G4 and G5 are the GTP binding regions, while G1–G3 are the GTPase activity center involved in GTP hydrolysis as well as phosphorylation and Mg^2+^ binding [12]. The subcellular localization of ROPs is controlled by HVR in the C terminus. The function of the insertion sequence of ROPs is currently unknown.

The Arabidopsis ROPs play distinctive yet overlapping roles during growth and development. For example, AtROP1 plays a role in promoting polar growth of pollen tubes, which is assisted by AtROP2, AtROP3, and AtROP6 [13]. Both AtROP2 and AtROP4 control the initiation, number, and density of root hairs as well as the elongation of the root tip [14]. Consistent with this, overexpressing AtROP2 and AtROP4 lead to the production of multiple swelling sites and the emergence of root hairs in the root expansion zone, resulting in an increased number and density of root hairs [15]. A similar function of AtROP6 was found where increased root hairs were the typical phenotype of the transgenic Arabidopsis plants [16].

ROPs are also involved in a variety of abiotic and biotic stress signals, such as drought, temperature (low and high), and diseases. For instance, *AtROP11* deletion mutant showed more drought resistance than plants expressing the constitutively active (CA) ROP11, demonstrating that AtROP11 is a negative regulator of the ABA-mediated stress response in Arabidopsis [17]. Under low temperature, the expression levels of *EjROP1.1*, *EjROP2,* and *EjROP3* in the fruits of Loquat (*Eriobotrya japonica* Lindl.) were upregulated. Interaction between the EjROPs, NADPH oxidase, and CCR (Cinnamoyl co-enzyme A reductase) was detected under cold treatment [18]. High temperature blocked the release of mature pollen grains from microspores in oilseed rape, which coincided with the altered expression levels of *BnROP5* and *BnROP9*, an indication of their important roles in the production of viable pollen grains [19]. ROP proteins are also involved in the disease response. One of the examples was the overexpression of *TaRac1* in tobacco where enhanced resistance to tobacco black shank and bacterial wilt diseases were observed due to the accumulation of total lignin and sinapyl lignin [20].

ROPs can be post-translationally modified for their proper function [21]. For example, AtROP2 is predicted to be S-acylated and this could be carried out by AtPAT4, one of the 24 protein S-acyltransferases of Arabidopsis [22,23,24].Importantly, S-acylation of AtROP2 plays an important role in its regulation of root hair polar growth in Arabidopsis [24]. However, very little is known whether other AtROPs in Arabidopsis or indeed ROPs in other plant species also go through this lipid modification and their roles in growth, development, and the stress response.

While the functions of a number of *ROPs* in Arabidopsis and other plant species have been well characterized, this is not so in trees, such as pear. Taking advantage of the recently sequenced genome of ‘*Duli’* pear (*Pyrus betulifolia* Bunge), a wild-type pear often used as rootstocks for grafting due to its resistance to abiotic and biotic stresses, and the important roles of *ROPs* in the regulation of growth, development, and stress responses in plants, we surveyed the genome data of ‘*Duli*’ pear and identified 12 *ROP* sequences, named as *PbROP1-12*. Subsequently, the phylogenetic relationship, gene structure, and conserved domains of their encoded proteins were analyzed. The temporal and spatial expression patterns as well as expression profiles of these ROPs upon treatments with ABA, PEG, and NaCl were monitored. Over-expression of 3 of the 12 *PbROPs* that are predicted to be S-acylated in Arabidopsis was also carried out and their phenotype upon ABA treatment was observed. These data could provide useful information for the systematic study of PbROPs and to guide molecular breeding of suitable rootstocks of ‘*Duli*’ pear in the future.

## 2. Results

### 2.1. Twelve PbROPs Were Identified from ‘Duli’ Pear Genome

To see if the ‘*Duli*’ pear genome also encodes ROPs, the amino acid sequences of the 11 well-characterized AtROPs of Arabidopsis were used to blast the genome database of ‘*Duli*’ pear [25,26]. This was followed by analyzing the conserved domains and motifs of the potential ROPs. Finally, 12 ROPs and their gene sequences were identified (Table 1). The numbers of nucleotides of the 12 PbROPs are highly variable, ranging from 1759 to 5549 bp, which encode proteins of 175 to 215 amino acids with Mw of 19.46 to 23.45 kDa and isoelectric points 8.71 to 9.83, respectively.

Phylogenetic analysis of the ROPs between ‘*Duli*’ pear and Arabidopsis was carried out. This showed that while Arabidopsis has all four types of ROPs, only three were found in ‘*Duli*’ pear, and type II ROP(s) are missing from its genome (Figure 1). The three types are type I (PbROP1-6, shaded orange), type III (PbROP8, PbROP12, shaded purple), and type IV (PbROP7, PbROP 9-11, shaded green), which all share high sequence similarity to the corresponding types in Arabidopsis. Further, an extra member was found in each of the type III and IV PbROPs compared to the same types of AtROPs in Arabidopsis.

*PbROPs* were positioned in chromosome 2 (*PbROP7* and *8*), 5 (*PbROP5* and *6*), 6 (*PbROP11*), 7 (*PbROP12*), 8 (*PbROP4*), 10 (*PbROP3*), 14 (*PbROP10*), and 15 (*PbROP1*, *2,* and *9*) (Figure 2). *PbROP5* and *6* were located on chromosome 5 with a sequence identity of 99.16%. Chromosome 15 had the most numbers of *PbROPs* with 3 members, *PbROP1*, *2,* and *9*.

### 2.2. The 12 PbROPs of ‘Duli’ Pear Are Typical ROPs of Plants

Multiple sequence alignment was performed to identify the conserved domains of the 12 newly identified ‘*Duli*’ pear PbROPs. Similar to Arabidopsis and other plant ROPs, 3 typical conserved functional domains were identified [10,11] from the deduced protein sequences of all 12 PbROPs, i.e., the G domain, insertion sequence, and HVR. The G1–G5 regions of the G domains for GTP binding and hydrolysis were also present (Figure 3). Therefore, the 12 PbROPs of ‘*Duli*’ pear are typical ROPs of plants based on their sequences.

### 2.3. The Gene Structure and Conserved Motifs Are Similar among the 12 PbROPs

*PbROPs* are predicted to have 7 to 8 exons (Figure 4b). Consistent with the number of exons of the type I and IV *ROPs* in Arabidopsis, *PbROP1-6*, *PbROP8,* and *PbROP12* also have 7 exons, while *PbROP7* and *PbROP9-11* in type III have 8 exons, which are the same as those in Type III *AtROPs* of Arabidopsis.

Seven conserved motifs were identified from the PbROPs (Figure 4c and Appendix A). Motifs 1-6 were found in all the 12 PbROPs, except PbROP4, which did not have Motif 3, while motif 7 was only found in PbROP10 and PbROP11. The conserved GTP-binding domain fell in motifs 1, 2, 3, and 5.

### 2.4. PbROPs Have Gone through Purifying Selection

The selection pressure of *PbROPs* was estimated (Appendix A). This showed that the ratios of Ka (synonymous mutations) / Ks (not synonymous mutations) of all *PbROPs* were less than 1, indicating that the *PbROP* family genes have been undergoing purifying selection [26].

The collinearity analysis conducted for gene fragment replication relationships put 10 of the 12 *PbROPs* into 5 pairs, i.e., *PbROP10*/*PbROP11*, *PbROP8*/*PbROP12, PbROP3*/*PbROP5, PbROP2*/*PbROP1,* and *PbROP2/ PbROP4* (Figure 5a). High sequence similarity was found between each pair of these *PbROPs*, indicating that the two *PbROPs* in a pair is conserved and may have similar functions during evolution.

To further understand the evolutionary relationship of the *PbROP* genes with those from Arabidopsis, syntenic and collinear regions of their *ROP* genes were analyzed by the Multiple Collinearity Scan toolkit. This resulted in the identification of many syntenic blocks (McScanX; Figure 5b). In total, 19 gene pairs were detected, indicating that the *ROPs* are evolutionarily conserved between ‘*Duli*’ pear and Arabidopsis (Appendix A).

### 2.5. Cis-Acting Elements Related to Growth and Development, and Hormonal and Stress Response Are Present in the Promoter Regions of PbROPs

The *cis*-acting elements within the promoter region are vital for the transcription and expression of genes, hence their functions [27]. Therefore, the ~2000 bp sequence upstream of transcriptional start codon ATG of all *PbROPs* was analyzed and the results are shown in Figure 6. In general, the *cis*-acting elements could be grouped into three types where type I is involved in growth and development, type II is involved in the hormonal response, and type III is involved in the stress response [28]. Within type I, there are 12 light response elements including 3 AF1 and 3 AF3 binding sites, MRE, G-box, lamp element, GT1-motif, GATA-motif, AE-box, BOX4, chs-CMA1a, I-box, and TCT-motif. Other type I *cis*-acting elements were also found, such as the GCN-motif (regulatory element required for endosperm expression), O_2_-site (*cis*-acting regulatory element involved in zein metabolism regulation), MBSI (MYB binding site involved in flavonoid biosynthetic genes regulation), HD-zip3 (leaf development correlated), MSA-like (involved in cell cycle regulation), CAT-box (involved in endosperm development), circadian (involved in meristem expression), and NON-box (meristem-specific activation). Within type II, the hormone response *cis*-acting elements, ABRE (ABA-responsive elements), CGTCA-motif (MEJA-responsive elements), ERE (ethylene-responsive elements), and TCA-element (salicylic acid-responsive elements), as well as TGA-elements, AuxRE related to the IAA response and gibberellin-responsive elements, TATC-box, P-box, and GARE-motif, were all present. Type III, the stress-responsive *cis*-acting elements, included ARE (involved in anaerobic induction), DRE (involved in dehydration, low-temperature, and high-salinity stresses), STRE (involved in stress), TC-rich repeats (involved in defense and stress), LTR (low temperature), WUN-motif (involved in wound), GC-motif (involved in anoxic specific inducibility), and MBS (involved in drought). ARE and STRE are shared by all *PbROPs*. Interestingly, *PbROP1* has 7 ARE elements, indicating that it may play a significant role in anaerobic induction.

### 2.6. Tissue-Specific Expression Pattern of PbROP Genes

In order to understand the biological functions of *PbROPs,* qRT-PCR was carried out to determine the temporal and spatial expression patterns of *PbROP* genes. The results in Figure 7 show that the expression levels of the type I *PbROPs*, *PbROP1-6,* were much higher in leaves and roots than other tissues, with the lowest levels found in fruits, indicating that these *PbROPs* play major roles in the growth and development of vegetative tissues. The expression levels of the *PbROP8* and *PbROP12*, the two type III *PbROPs*, were similar where they were much higher in roots than other tissues, implying that they are mainly involved in the regulation of root growth and development. The type IV *PbROPs* have a more complicated expression pattern where the highest expression levels of *PbROP10* and *11* were found in roots, and those of *PbROP7* and *PbROP9* were found in leaves and fruits, respectively.

### 2.7. Expression Levels of PbROPs Were Elevated upon Stress

Tissue-cultured seedlings of ‘*Duli*’ pear were treated with ABA, PEG, and NaCl to mimic overall abiotic stress, drought, and salt stress, respectively. The transcript level of each *PbROP* was quantified by real-time RT-PCR using total RNAs isolated from the treated seedlings collected after various time points. This showed that all the 12 *PbROPs* were upregulated to different degrees when treated with 1 µM ABA where the highest expression levels of 7 *PbROPs* (*PbROP2, 3, 4, 7, 8, 9,* and *12*) were reached at 72 h, while those of *PbROP1* and *10* were at 120 h (Figure 8). It is noteworthy that the expression level of PbROP12 increased 14-fold after 72 h compared to 0 h. Upon treatment with 150 mM of NaCl, the transcript levels of *PbROPs* first increased and then decreased, where 7 *PbROPs (PbROP1, 2, 3, 4, 6, 10,* and *11)* reached the highest at 24 h, whilst those of *PbROP7, 8, 9,* and *12* and *PbROP5* were at 72 h and 12 h, respectively (Figure 8). Similarly, the expression levels of *PbROPs* in PEG-treated seedlings first increased and then decreased, where *PbROP3* and *PbROP6* showed the highest expression levels at 3 h; *PbROP8, 10,* and *11* at 6 h; *PbROP1, 2, 4, 7,* and *9* at 12 h; and *PbROP5* and *12* at 24 h (Figure 8). 

### 2.8. PbROP1, 2, 3, and 9 Overexpressing Transgenic Arabidopsis Seedlings Have Different Sensitivity to ABA

Seeds of the four individual *PbROP**s* overexpressing homozygous transgenic Arabidopsis plants were germinated on ½ MS with or without 0.5 μM ABA. After 10 days, the percentage of seedlings with green cotyledons were calculated. As shown in Figure 9, while the number of seedlings with green cotyledons were 56.7% in WT, this was much reduced in the *PbROP1, 2,* and *9* transgenic Arabidopsis seedlings where the percentages of seedlings with green cotyledons of the 3 lines of *PbROP1* over-expressers were 22.7%, 22.7%, and 22.0%, those of the three *PbROP2* transgenic lines were 29.8%, 25.5%, and 19.2%, while only 16.3%, 15.6%, and 17.0% of seedlings with green cotyledons were found in the 3 lines of *PbROP11* transgenic seedlings. However, 0.5 μM ABA had little effect on the percentage of seedlings with green cotyledons of the transgenic Arabidopsis seedlings expressing *PbROP3,* where 58.2%, 57.5%, and 56.7% of green seedlings were recovered, which were very similar to the ABA-treated WT seedlings. Therefore, the fact that *PbROP1,*
*2,* and *9* transgenic lines are more sensitive to ABA than *PbROP**3* transgenics suggests that *PbROP1,*
*2,* and *9* may play specific roles in ABA signal regulation.

## 3. Discussion

Because of its resistance to drought, flood, salinity, and diseases, ‘*Duli*’ pear (*Pyrus betulifolia* Bunge) is used as the main rootstock for pear planting in north China [29]. However, the ‘*Duli*’ pear plant is naturally tall with well-developed taproot roots, and fewer and weaker lateral roots in the seedling stage. It is also difficult in rooting from cutting. However, as desired rootstocks, dwarf and easy rooting are the required characteristics. Previous studies of ROPs found that they play important roles in root hair development, plant growth, and stress tolerance [3,7,30]. Therefore, research on *ROPs* of ‘*Duli*’ pear is of great significance not only in revealing their roles in the development and stress resistance mechanism of woody plants, but also providing information on candidate genes to use in molecular breeding for improved rootstock for the pear industry.

In this study, we identified 12 ROPs from the genome of ‘*Duli*’ pear, which are more than those of Arabidopsis (11) [3] and Rice (7) [4] (Figure 1, Table 1). Further analysis demonstrated that the increased number may be caused by the gene expansion events during evolution. One homologous gene pair, *PbROP5*/*PbROP6,* and four paralogous pairs (*PbROP2*/*PbROP4*, *PbROP7*/*PbROP9*, *PbROP8*/*PbROP12,* and *PbROP10*/*PbROP11*) were found in ‘*Duli*’ pear *PbROPs*. They are most likely originated from genome-wide chromosome replication events during evolution (Figure 2).

The analysis of gene structure, phylogeny, and amino acid sequence similarity showed that the members of the ROP family were much conserved in structure and amino acid sequence (Figure 3 and Figure 4A). According to the amino acid sequence similarity and the structure of the highly variable C-terminal regions, the *PbROPs* were divided into three types rather than four in Arabidopsis because type II, i.e., the AtROP7 homologous gene, is missing from ‘*Duli*’ pear (Figure 1). A study in Arabidopsis showed that AtROP7 and AtROP2 are functionally redundant; together, they play negative roles in controlling light-induced stomatal opening [31]. However, how and what the implication is for the ‘*Duli*’ pear genome to lack the type II ROP remains to be determined.

Many studies of plant *ROPs* showed that they are involved in plant growth and development as well as the stress response. For example, *AtROP2* [32] and *MaROP5* [7] participate in the response to salt stress, and *EjROPs* are related to cold stress [18]. To see if PbROPs play similar roles, we first analyzed the promoter regions of the 12 *PbROPs*. This revealed that the *cis*-acting elements related to growth and development, and the phytohormone and stress response are all present (Figure 6). This indicates that the 12 *PbROPs* may also function in growth, development, and abiotic stress in ‘*Duli*’ pear. For example, the promoter regions of all 12 *PbROPs* contain the stress-responsive elements ARE and STRE. It is interesting that *PbROP3* also contains six lower-temperature-responding (LTR) elements, indicating it may play a significant role in the regulation of low-temperature stress. We also identified *cis*-acting elements related to ABA, ethylene, IAA, GA, MeJA, and SA signaling, indicating that PbROPs may also be involved in many hormone signal regulation processes during growth and development of the ‘*Duli*’ pear plant.

Further analysis of the expression profiles of the *PbROPs* in different tissues of mature plants as well as in tissue-cultured seedlings treated with ABA, NaCl, and PEG provided more evidence to support the bioinformatic data in general. For example, we found that type I and Ⅳ *PbROPs* were expressed in roots to high levels, indicating that these two types of *PbROPs* may play an important role in root growth. Similar findings were also made with the *AtROPs* in Arabidopsis [33,34,35]. As a stress hormone, ABA plays crucial roles in plant responses to abiotic stress [17]. Transgenic Arabidopsis plants expressing *AtROP2* and *AtROP11* exhibited altered ABA responses [3,17]. Our study also showed that the expression levels of the majority of *PbROPs* were increased in ‘*Duli*’ pear seedlings treated with 1 µM of ABA (Figure 8). Further, transgenic Arabidopsis expressing *PbROP1, 2,* or *9* became super sensitive to ABA because the survival rate of seedlings (with green cotyledons) was reduced from 56.7% for WT to 22.5%, 24.8%, and 16.3% for 35S:PbROP1, 35S:PbROP2, and 35S:PbROP9 transgenic seedlings, respectively, while that of *PbROP3* expressing Arabidopsis was very similar to the WT (Figure 9). It is worth noting that one common feature shared by these four PbROPs is that they are all predicted to be S-acylated at Cys9. Given the important roles of S-acylation in membrane targeting, which is directly related to the functionality of ROPs, this warrants further investigation in order to understand if/how S-acylation of these four PbROPs occurs in ABA signaling and stress of ‘*Duli*’ pear [21].

When ‘*Duli*’ seedlings were treated with NaCl and PEG, the expression levels of *PbROPs* were increased and then decreased, indicating that these genes may participate in the initial stage of response to these stress conditions. In tobacco, the expression level of *NtROP1* was increased upon treatment with NaCl. When *NtROP1* was expressed in Arabidopsis, the transgenic plants showed increased salt sensitivity [36]. A study in Arabidopsis showed that the involvement of AtROP2 in the salt response is mediated by its involvement of the reassembly of microtubules where the constitutively active ROP2 promoted the reassembly of microtubules and the survival of seedlings, while the *rop2-1* mutant showed a significantly low survival rate under salt stress [32].

The drought signal can excite the plant response to ABA, BR, ethylene, and ROP proteins [37,38]. Arabidopsis At*ROP11* exerts its role in drought stress by playing a negative regulation role in the closure of stomata induced by ABA [17]. HDA15 (Histone modifier 15) loss-of-function mutant *hap15* is sensitive to drought. This is because the MYB96-HDA15 complex binds to the promoters of a subset of ROPs, including ROP6, ROP10, and ROP11, to regulate their expression [39]. The fact that ABA can enhance the expression of PbROPs in ‘*Duli*’ pear seedlings and reduce the survival rates of transgenic Arabidopsis seedlings overexpressing three of the PbROPs indicated that the PbROPs may play important roles in the stress response via similar molecular mechanisms. Therefore, PbROPs may be crucial in the adaptation and survival of ‘*Duli*’ pear under abiotic stress conditions.

## 4. Materials and Methods

### 4.1. Data Acquisition

To build a local database, the reference sequence of the ‘*Duli*’ pear genome was downloaded from the database of the Chinese Academy of Sciences [26]. Using the sequences of *AtROPs* of Arabidopsis and Tbtools, the ‘*Duli’* pear genome was blasted to identify homologous *ROP* gene sequences (*E-value* < 10^−10^) [40]. This was followed by analyzing the conserved domains of the putative PbROP amino acid sequences using NCBI-CDD (https://www.ncbi.nlm.nih.gov/cdd/ (accessed on 2 March 2022)) and ExPAsy (https://www.expasy.org/ (accessed on 5 March 2022)) [41].

### 4.2. Phylogenetic Tree, Gene Structure, and Sequence Analysis of PbROP Gene Family

Clustal (http://www.clustal.org/ (accessed on 8 March 2022)) was used to compare the amino acid sequences of the ROP family of ‘*Duli*’ pear. Sequence similarity analysis was calculated using Genedoc (http://www.psc.edu/biomed/genedoc/ (accessed on 12 March 2022)). MEGA6.0 software(version 6.0, Mega Limited, Auckland, New Zealand) was used to construct the phylogenetic tree by the Neighbor-Joining (NJ) method with bootstrap 1000 and other default parameters [42]. The gene structure was constructed using Tbtools [40].

### 4.3. Chromosome Location, Conserved Motifs, and the Cis-Elements in the PROMOTERS of PbROPs of ‘Duli’ Pear

The specific positions of each of the 12 *PbROPs* on the 17 chromosomes were mapped and drawn using Tbtools [40]. The online software MEME (http://meme-suite.org/meme/, accessed on 18 March 2022) was used to identify the conserved motifs of the PbROPs with the parameters set to 20 and the rest as default (https://meme-suite.org/meme/doc/meme.html (accessed on 22 March 2022)) [43]. The important *cis*-elements 2000 bp upstream of the transcriptional initiation codon ATG of the promoter regions of each *ROP* were identified using the PlantCARE online software (http://bioinformatics.psb.ugent.be/webtools/plantcare/html/ (accessed on 22 March 2022)) [44].

### 4.4. Selective Pressure and Collinearity Analysis of PbROPs

According to the gene clustering of the ROP families, the collinearity of the genes between Arabidopsis and ‘*Duli*’ pear were compared with Muscle built into MEGA6.0. The selection pressure between ROP genes was calculated by DNAsp6.12.01 (DNA Sequence Polymorphism). The collinearity of ROP genes of ‘*Duli*’ pear and Arabidopsis was analyzed by McScanX (http://chibba.pgml.uga.edu/mcscan2/ (accessed on 24 March 2022)), and the relatedness diagram was drawn by Tbtools [40].

### 4.5. Expression Patterns of PbROPs

Three 5-year-old ‘*Duli*’ pear plants with similar growth vigor were selected from the pear resources nursery of the Pear Engineering Technology Research Center of Hebei Agricultural University, China. Roots, shoots, leaves, flowers, and young fruits were collected for RNA isolation (see below).

### 4.6. Expression Profiles of PbROPs in Response to Different Stress Treatments

Tissue-cultured ‘*Duli*’ pear seedlings were grown in the plant tissue culture room under a long day of 16 h light and 8 h dark at a temperature of 24 ± 2 °C. The control seedlings were cultured in the base MS medium (MS 4.4 g/L + 6-BA 1.0 mg/L + IBA 0.1 mg/L + sucrose 30 g/L + Agar 8 g/L pH 5.8), while 1.0 μM ABA, 150 mM NaCl, or 20% PEG6000 were added to the base MS media for different stress treatments. Samples were collected at 0, 3, 6, 12, 24, 72, and 120 h after each treatment and immediately frozen in liquid nitrogen and stored at −80 °C until further analysis by RT-PCR (see below).

### 4.7. Total RNA Extraction and Synthesis of 1st-Strand cDNAs

Total RNA was extracted from the above collected tissues of ‘*Duli*’ pear plants using the TIANGEN RNA prep Pure Plant Kit (Tiangen Biotech, Beijing, China) according to the manufacturer’s protocol. The quality of the RNA was verified by gel electrophoresis and quantified using Nanodrop. One microgram of total RNA was reverse-transcribed to the first-stranded cDNA following the method provided by the TIANGEN FastKing cDNA first-strand synthesis kit (Tiangen Biotech, Beijing, China).

### 4.8. Real-Time PCR

The primers for quantitative real-time PCR were designed using Primer 5.0 and are listed in Appendix A. A total of 20 μL of PCR mix containing 10 μL of SuperReal Premix Plus (TransGen Biotech, Beijing, China), 600 ng of cDNA, and 0.5 μL of each forward and reverse primer (10 µM) were set-up. The LightCycler 96 real-time PCR machine (Roche, Basel, Swiss Confederation) was used to run the PCR reaction with the following regime: pre-denaturation at 94 °C for 30 s, 45 cycles of 94 °C for 5 s, 55 °C for 10 s, and 72 °C for 10 s. The *β-actin* gene was used as the reference gene and the relative expression level of each *PbROP* was calculated. Three biological replicates and three technical replicates were included for each gene.

### 4.9. Over-Expression of PbROPs and ABA Sensitivity Assay of Transgenic Arabidopsis Seedlings

The full coding regions of *PbROP1*, *PbROP2*, *PbROP**3,* and *PbROP9* were individually cloned using the primers list in Appendix A into the plant expression vector pEarleyGate103 via the Gateway cloning strategy [45,46]. After transforming *Agrobacterium* strain GV3101, the floral dipping method was used to transform Arabidopsis [47]. The homozygous transgenic lines were selected in T_3_ progenies. Forty seeds of each overexpression homozygous line were germinated on ½ MS medium containing 0, 0.5, and 1 μM ABA, respectively. They were cultured under a long day with 16 h light/8 h dark at 25 ± 1 °C. After 10 days, the seedlings were scanned and the seedlings with green cotyledons were scored. Each treatment was repeated three times for statistical analysis.

## 5. Conclusions

In summary, we identified 12 PbROPs from the wild-type ‘*Duli*’ pear. They are typical ROPs because they share sequence similarity and have all the conserved domains and motifs similar to ROPs from Arabidopsis and other plant species. Expression analysis by qRT-PCR of all 12 PbROPs and transgenic studies of 4 PbROPs in Arabidopsis indicated that they play important roles in growth, development, and particularly in abiotic stress. Further research will focus on the molecular mechanisms, and importantly, the involvement of S-acylation in the regulation of biological function of PbROPs in ‘*Duli*’ pear.

## Figures and Tables

**Figure 1 plants-11-01608-f001:**
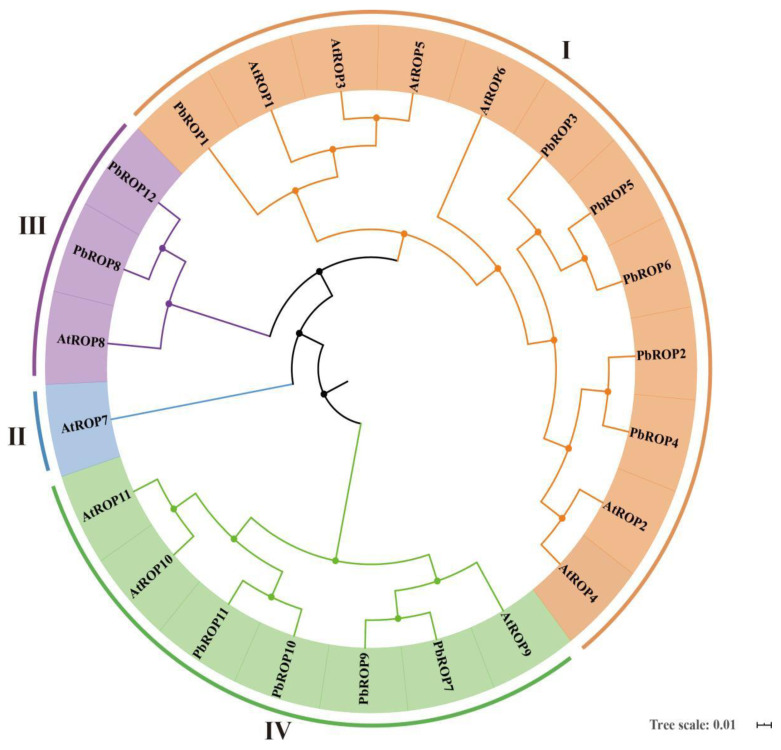
Phylogenetic analysis of ROPs of **‘***Duli*’ pear (*Pyrus betulifolia* Bunge) and Arabidopsis. The 4 types of ROPs were marked with different colors, as indicated in the graph. The phylogenetic tree was constructed using the neighbor-joining (NJ) method for alignment and 1000 bootstrap replications.

**Figure 2 plants-11-01608-f002:**
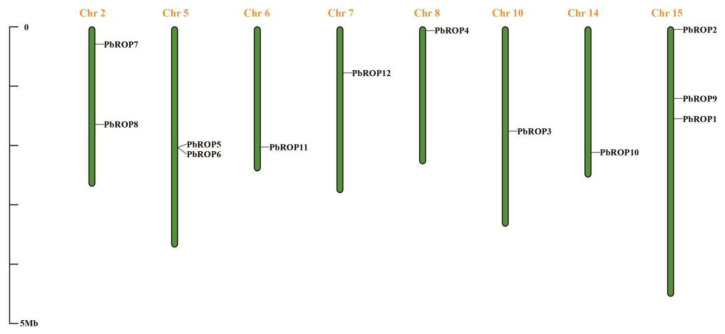
The distribution of *PbROPs* on different chromosomes of ‘*Duli*’ pear (*Pyrus betulifolia* Bunge). The chromosome number is indicated on the top of each chromosome. Base-pair positions are indicated on the left vertical line (0–5 Mb).

**Figure 3 plants-11-01608-f003:**
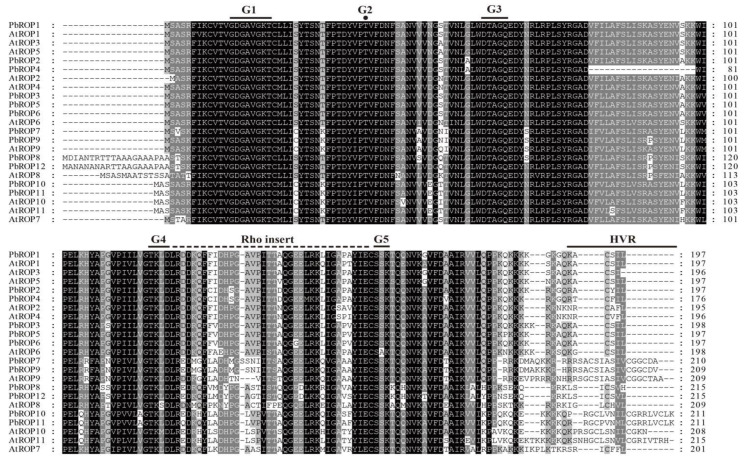
Multiple alignment of the amino acid sequences of PbROPs of ‘*Duli*’ pear (*Pyrus betulifolia* Bunge) and AtROPs of Arabidopsis. The G1–G5 of the G domain, Rho insert (Rho insertion sequence), and HVR (the hypervariable region) are indicated.

**Figure 4 plants-11-01608-f004:**
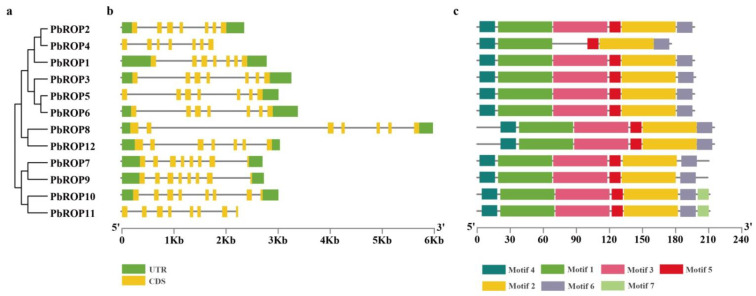
Phylogenetic relationships, gene structure, and conserved motifs of the encoded proteins of *PbROPs* of ‘*Duli*’ pear (*Pyrus betulifolia* Bunge). (**a**) The phylogenetic tree. It was constructed based on the full-length PbROP protein sequences using MEGA 5.0 software. (**b**) Gene structure of *PbROPs*. Green boxes indicate 5′- and 3′-untranslated regions, yellow boxes are exons, and black lines are introns. The length of each *PbROP* can be estimated using the scale below the figure. (**c**) The conserved motifs and their positions of PbROP proteins. The sequence information of each motif is provided in Appendix A. The number of amino acids in each motif can be estimated using the scale below the figure.

**Figure 5 plants-11-01608-f005:**
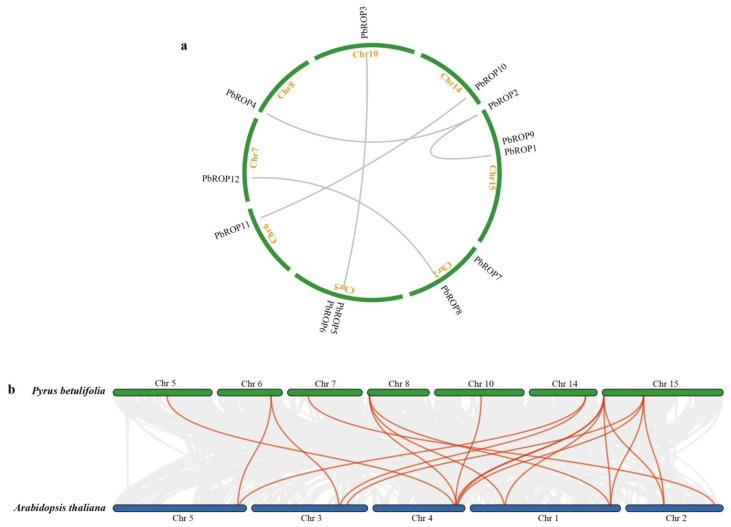
Inter-chromosomal relationship of *PbROPs* of ‘*Duli*’ pear (*Pyrus betulifolia* Bunge) and synteny analysis of *PbROPs* and *AtROPs* of Arabidopsis. (**a**) Schematic representations of the chromosomal distribution and inter-chromosomal relationships of *PbROPs* of ‘*Duli*’ pear. Gray lines indicate duplicated *ROP* genes. The chromosomes are presented as green lines, which are labeled with orange letters. (**b**) Synteny analysis between *Pb**ROPs* of ‘*Duli*’ pear and *AtROPs* of Arabidopsis. Gray lines in the background indicate the collinear blocks that are highlighted in red lines to indicate the syntenic *ROP* gene pairs between *Pb**ROPs* and *AtROPs*.

**Figure 6 plants-11-01608-f006:**
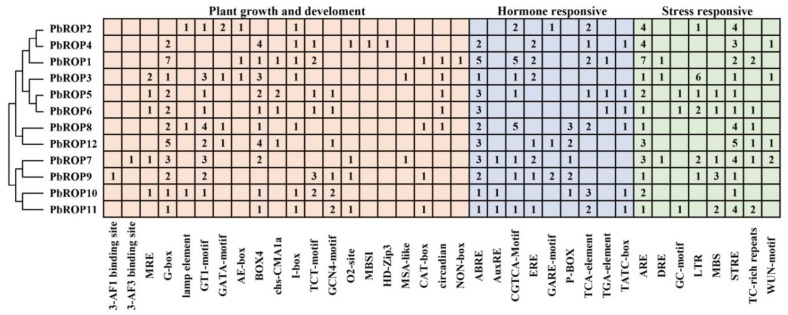
Analysis of the *cis*-acting elements in the promoter regions of *PbROP* genes of ‘*Duli*’ pear (*Pyrus betulifolia* Bunge). The three common *cis*-acting elements identified were related to (1) plant growth and development (shaded in pink), (2) hormone response (shaded blue), and (3) stress response (shaded green). The number represents how many copies of *cis*-acting elements are present. The 2 kb sequences upstream of the transcriptional start codon ATG of all 12 PbROP genes were analyzed using the PlantCARE software. The figure was constructed by Microsoft and MEGA 6.0 software.

**Figure 7 plants-11-01608-f007:**
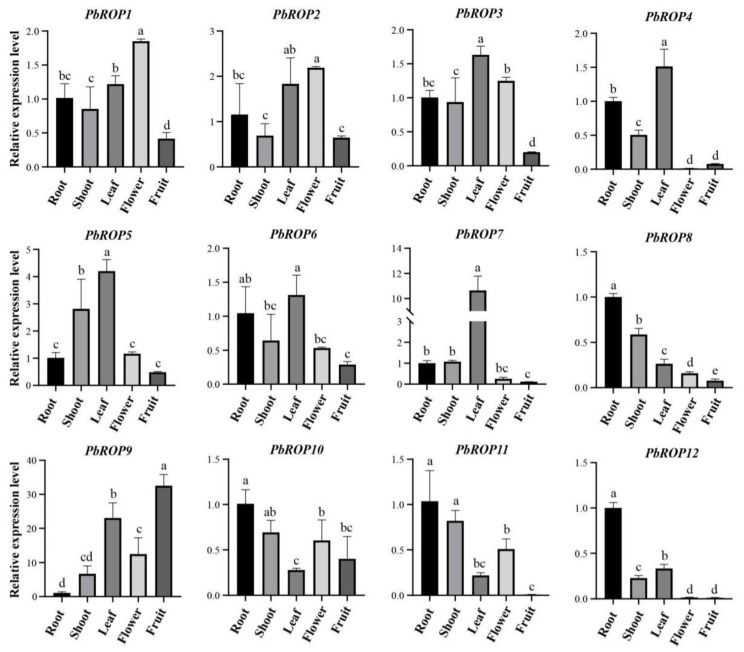
Expression profiles of *PbROPs* in different tissues of ‘*Duli*’ pear monitored by qRT-PCR. The expression levels of the 12 *PbROPs* in roots, shoots, leaves, flowers, and young fruits were measured by real-time RT-PCR. *PbACTIN* was used as the reference gene. The relative expression level of each gene was calculated relative to the transcript level of the roots (set as 1). Each bar represents the mean ± SE (n = 3). The chart was constructed using GraphPad Prism 8.0 software. Different letters on top of the columns indicate significant differences at *p* < 0.05 (n = 3) using Duncan’s multiple range test by the SPSS software.

**Figure 8 plants-11-01608-f008:**
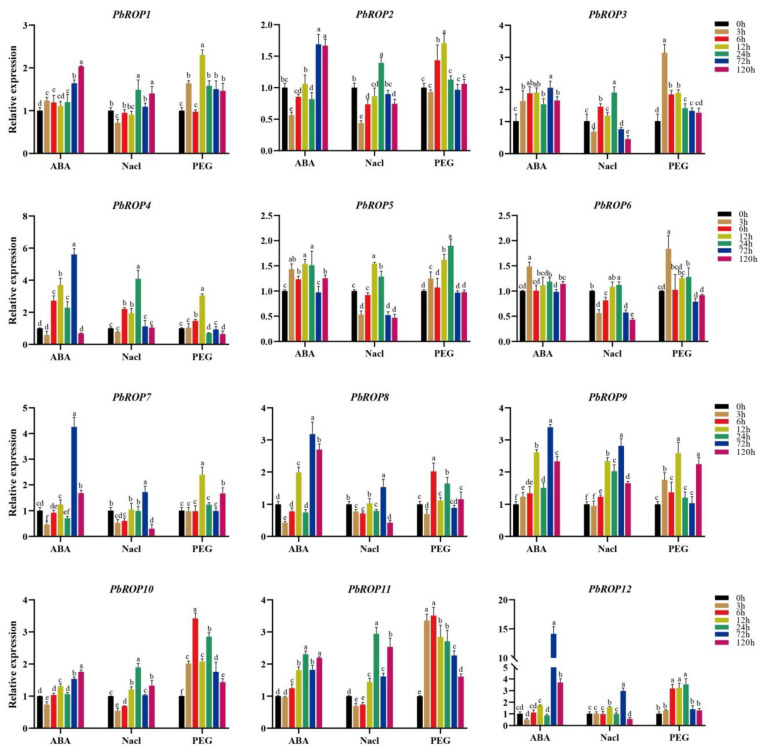
Expression profiles of *PbROPs* in tissue-cultured ‘*Duli*’ pear seedlings treated with 1 μM ABA, 150 mM NaCl, and 20% PEG. The relative expression levels of *PbROPs* at 5 different time points were monitored by real-time RT-PCR. *PbACTIN* was used as the reference gene. The relative expression level of each gene was calculated relative to the transcript level at the start of treatment (0 h) (set as 1). Each bar represents the mean ± SE (n = 3). The chart was constructed using GraphPad Prism (8.0.2, Harvey Motulsky, California, USA) software. Different letters on top of the columns indicate significant differences at *p* < 0.05 (n = 3) using Duncan’s multiple range test by the SPSS (25.0, IBM company, New York, NY, USA) software.

**Figure 9 plants-11-01608-f009:**
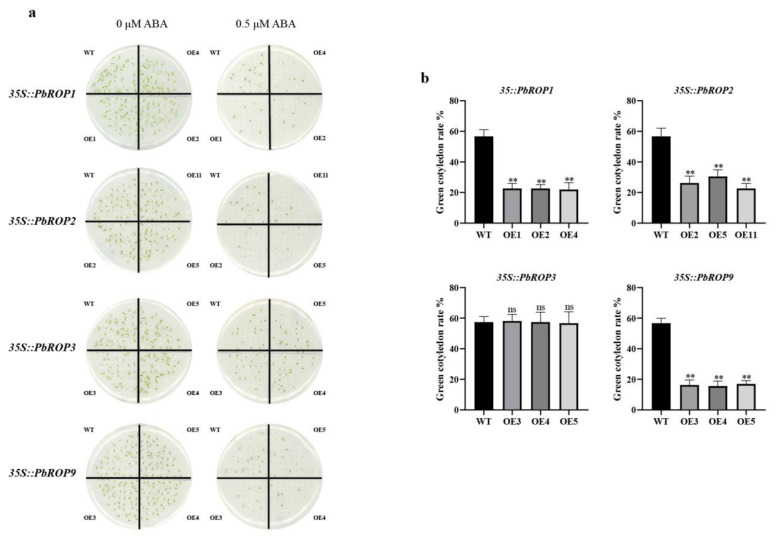
*PbROP1, 2, 3,* and *9* overexpressing transgenic Arabidopsis seedlings have different sensitivity to ABA. (**a**) The phenotype of the *35S::PbROPs* Arabidopsis seedlings grown on plates without or with 0.5 µM ABA; (**b**) percentage of seedlings with green cotyledons of the *35S::PbROPs* transgenic Arabidopsis seedlings grown on plates without or with 0.5 µM of ABA. Seeds of homozygous transgenic Arabidopsis were surface-sterilized and germinated on ½ MS with or without 0.5 µM ABA. After 10 days, the phenotype was observed and the percentage of seedlings with green cotyledons was calculated. The values represent the means ± SE (n = 3). The data were analyzed via the Student’s *t*-test using GraphPad Prism (8.0.2, Harvey Motulsky, California, USA) software. ** indicate significant differences in comparison with WT at *p* < 0.01. ns, no significant difference.

**Table 1 plants-11-01608-t001:** Characteristics of the 12 *PbROP* genes of ‘*Duli*’ pear (*Pyrus betulifolia* Bunge).

Name	Gene Acc Number	Gene ID	Gene Location	Length of Gene (bp)	No. Amino Acid	Mol Weight (kDa)	Protein Isoelectric Point/(pI)
PbROP1	GWHPAAY-T021718	Chr15.g02695.m2	Chr15(−)15491640-15493489	1850	197	21.55	9.7
PbROP2	GWHPAAYT019557	Chr15.g04856.m1	Chr15(+)449034-450843	1809	197	21.74	9.34
PbROP3	GWHPAAYT003833	Chr10.g15874.m1	Chr10(+)17567259-17569896	2638	198	21.83	9.82
PbROP4	GWHPAAYT050667	Chr8.g53635.m1	Chr8(+)644981-646739	1759	176	19.46	8.71
PbROP5	GWHPAAYT042018	Chr5.g07968.m1	Chr5(+)20278696-20281398	2703	197	21.65	9.7
PbROP6	GWHPAAYT042034	Chr5.g07952.m1	Chr5(+)20386888-20389606	2719	197	21.60	9.83
PbROP7	GWHPAAYT031525	Chr2.g41544.m1	Chr2(+)2929720-2931810	2091	210	23.12	9.25
PbROP8	GWHPAAYT033120	Chr2.g43139.m1	Chr2(−)16443373-16448921	5549	215	23.45	9.33
PbROP9	GWHPAAYT021263	Chr15.g03150.m1	Chr15(−)12072153-12074312	2160	209	23.07	9.24
PbROP10	GWHPAAYT018785	Chr14.g49938.m2	Chr14(+)21184845-21187330	2486	211	23.39	9.54
PbROP11	GWHPAAYT046524	Chr6.g51364.m1	Chr6(+)20257138-20259368	2231	211	23.42	9.81
PbROP12	GWHPAAYT048105	Chr7.g33937.m1	Chr7(+)7754549-7757181	2633	215	23.42	9.08

## Data Availability

Not applicable.

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
