# Peer review of "Genome-Wide Identification of Genes Encoding for Rho-Related Proteins in ‘Duli’ Pear (Pyrus betulifolia Bunge) and Their Expression Analysis in Response to Abiotic Stress"

_plants, 2022, doi:10.3390/plants11121608_

Round 1

Reviewer 1 Report

This manuscript “Genome-wide identification and expression analysis of PbROP gene family in response to multiple stresses in ‘Duli’ pear (Pyrus betulifolia Bunge)” is an interesting paper and should be published. However, before publishing it, I suggest you pay attention to a few things:

  1. Why were only Arabidopsis plants selected for comparison? In the discussion, too, the authors confine themselves to comparing the ‘Dula’ pear with Arabidopsis. Is this research not comparable to research conducted for other plants?
  2. Figure 8 is difficult to read. You can enlarge the charts or make colored bars then the charts would be more readable.
  3. I suggest that you read the manuscript carefully in order to remove minor errors, such as punctuation.
  4. Lack of conclusions or any summary of the conducted research and looking into the future.

Reviewer 2 Report

The paper is written in a satisfactory English and scientific style, moderate corrections are needed and they would better be provided by professionals. The authors present an extensive characterization of the 12 ROP genes from Duli pear genome.  Genomic analysis is flanked by expression profiling in plant tissues and in seedlings under stress conditions. In addition functional characterization is also conducted via over-expression in Arabidopsis and analysis of seedlings ABA sensitivity.

Line 230 reports that seedlings were treated with 1 microM ABA for the expression analysis under stress, the relative Figure 8 legend does not report the ABA concentration (please input it and also the concentration of the other stressors for clarity). The materials and methods at line 389 report 0,5 microM ABA used for the stress treatment, please clarify.

Concerning the over-expression experiment line 247 makes you believe that all the 12 ROP genes were over-expressed, please re-phrase. Also please state why only 4 genes were chosen for the overexpression experiment (based on the expression analysis under stress? but most of the genes are modulated..).

The germination rate of Columbia under 0,5 microM ABA in the authors experiment is low compared to what is found in the literature (see papers such as www.plantphysiol.org/cgi/doi/10.1104/pp.106.081018, doi: 10.1093/mp/ssr030) that report something around 70% germination rate (also as seedlings having green cotyledons). In addition the authors have different germination rate for Columbia in plates related to different genes (47.5%, 15%, 31,7 %) although the plates should have a similar composition. Please comment about it.

The seedlings of the over-expressing lines were subjected only to ABA treatment, as the authors have provided evidence of expression modulation also in case of salt and PEG treatment, it could be interesting to use also the other two stressors.

line 293-294 too much speculation about the taproots and the genomic condition of the two ROP genes, please remove.

lines 301-302 are clearly pasted from somewhere and with two mistakes

Round 2

Reviewer 1 Report

The manuscript has been sufficiently revised and is suitable for publication in the current version.

Author Response

Thank you very much.

Reviewer 2 Report

First of all the language was not improved, it is very striking to still see the "for example" at the end of the abstract..

Point 2: Concerning the over-expression experiment line 247 makes you believe that all the 12 ROP genes were over-expressed, please re-phrase. Also please state why only 4 genes were chosen for the overexpression experiment (based on the expression analysis under stress? but most of the genes are modulated..).

Response 2: We have changed to the subheading to reflect the ones overexpressed in Arabidopsis as "Transgenic Arabidopsis seedlings overexpressing PbROP1, 2&9 are sensitive to ABA". The reason to choose these PbROPs was because they were predicted to be palmitoylated. Given the main focus of our research group on protein palmitoylation we will explore further the relationship between palmitoylation of ROPs and abiotic stress in pear in the future.

reply: The choice of the genes to over-express based on palmitoylation is not very clear from the text, it should be stated either at the end of the introduction where the results are briefly summarized or at the beginning of the relative results section. The relative section starts with: Seeds of the individual PbROP overexpressing homozygous transgenic Arabidopsis plants were germinated on ½ MS with or without 0.5μM ABA etc.. The reader wonders which individual transgenics?? which genes and why?

Point 3: The germination rate of Columbia under 0,5 microM ABA in the authors experiment is low compared to what is found in the literature (see papers such as www.plantphysiol.org/cgi/doi/10.1104/pp.106.081018, doi: 10.1093/mp/ssr030) that report something around 70% germination rate (also as seedlings having green cotyledons). In addition the authors have different germination rate for Columbia in plates related to different genes (47.5%, 15%, 31,7 %) although the plates should have a similar composition. Please comment about it.

Response 3: We observed and recorded the % of green seedlings after 7 days of seed germination instead of 10 days used in the mentioned literature, hence the lower germination rate reported here. The variation in germination rates of Col-0 for different genes were indeed observed in this study. This could be due to the slight different culture conditions such as temperature and light that the different plates were placed under. However, there was no significant difference between the three replicates for Col-0. Therefore, this should bear no advert effect on the conclusion drawn for the response of each gene to ABA treatment.

reply: I understand the different germination rate compared to the literature which is due to the authors scoring time (but why not to score as what is commonly reported in literature?). The use of 1 microM ABA should be commented in the relative results section (as the picture in Fig.9a concerning plates at 1 microM ABA is still there) or remove the picture with 1 microM ABA. Considering the different germination rate of Columbia for the 4 over-expressing lines it would be appropriate to repeat the test with all the plates in the same condition.

Round 3

Reviewer 2 Report

The paper is substantially improved.